# Towards Neural Theorem Proving at Scale

## Abstract

Neural models combining representation learning and reasoning in an end-to-end trainable manner are receiving increasing interest. However, their use is severely limited by their computational complexity, which renders them unusable on real world datasets. We focus on the Neural Theorem Prover (NTP) model proposed by Rocktäschel and Riedel (2017), a continuous relaxation of the Prolog backward chaining algorithm where unification between terms is replaced by the similarity between their embedding representations. For answering a given query, this model needs to consider all possible proof paths, and then aggregate results – this quickly becomes infeasible even for small Knowledge Bases (KBs). We observe that we can accurately approximate the inference process in this model by considering only proof paths associated with the highest proof scores. This enables inference and learning on previously impracticable KBs.

## 1. Introduction

Recent advancements in deep learning intensified the long-standing interests in integrating symbolic reasoning with connectionist models (Shen, 1988; Ding et al., 1996; Garcez et al., 2012; Marcus, 2018). The attraction of said integration stems from the complementing properties of these systems. Symbolic reasoning models offer interpretability, efficient generalisation from a small number of examples, and the ability to leverage knowledge provided by an expert. However, these systems are unable to handle ambiguous and noisy high-dimensional data such as sensory inputs (Raedt and Kersting, 2008). On the other hand, representation learning models exhibit robustness to noise and ambiguity, can learn task-specific representations, and achieve state-of-the-art results on a wide variety of tasks (Bengio et al.,

2013). However, being universal function approximators, these models require vast amounts of training data and are treated as non-interpretable *black boxes*.

One way of integrating the symbolic and sub-symbolic models is by continuously relaxing discrete operations and implementing them in a connectionist framework. Recent approaches in this direction focused on learning algorithmic behaviour without the explicit symbolic representations of a program (Graves et al., 2014; 2016; Kaiser and Sutskever, 2016; Neelakantan et al., 2016; Andrychowicz and Kurach, 2016), and consequently with it (Reed and de Freitas, 2016; Bosnjak et al., 2017; Gaunt et al., 2016; Parisotto et al., 2016). In the inductive logic programming setting, two new models, NTPs (Rocktäschel and Riedel, 2017) and Differentiable Inductive Logic Programming ($\partial$ILP) (Evans and Grefenstette, 2018) successfully combined the interpretability and data efficiency of a logic programming system with the expressiveness and robustness of neural networks.

In this paper, we focus on the NTP model proposed by Rocktäschel and Riedel (2017). Akin to recent neural-symbolic models, NTPs rely on a continuous relaxation of a discrete algorithm, operating over the sub-symbolic representations. In this case, the algorithm is an analogue to Prolog's backward chaining with a relaxed unification operator. The backward chaining algorithm constructs neural networks, which model continuously relaxed proof paths using sub-symbolic representations. These representations are learned end-to-end by maximising the proof scores of facts in the KB, while minimising the score of facts not in the KB, in a link prediction setting (Nickel et al., 2016). However, while the symbolic unification checks whether two terms can represent the same structure, the relaxed unification measures the similarity between their sub-symbolic representations.

This continuous relaxation is at the crux of NTPs' inability to scale to large datasets. During both training and inference, NTPs need to compute all possible proof trees needed for proving a query, relying on the continuous unification of the query with *all* the rules and facts in the KB. This procedure quickly becomes infeasible for large datasets, as the depth and width of the resulting computation graph grow exponentially.

Our insight is that we can radically reduce the computational complexity of inference and learning by generating only the

[1]Anonymous Institution, Anonymous City, Anonymous Region, Anonymous Country. Correspondence to: Anonymous Author <anon.email@domain.com>.

Preliminary work. Under review by the International Conference on Machine Learning (ICML). Do not distribute.

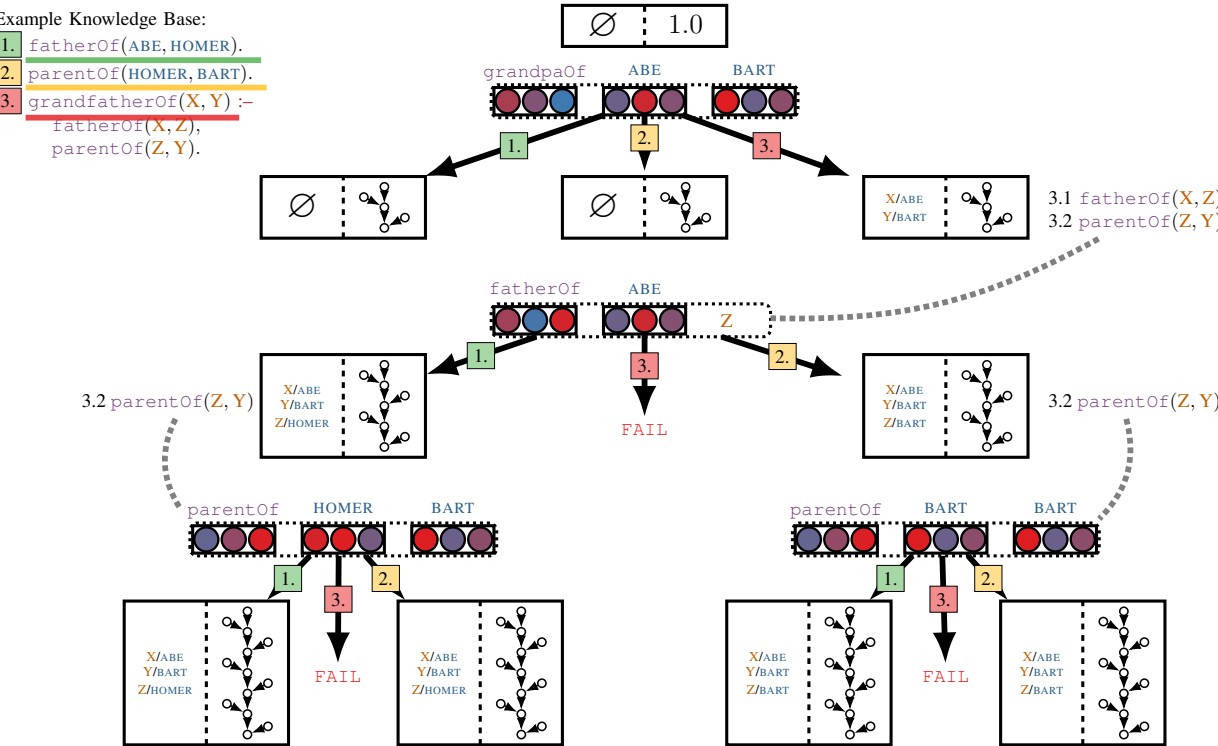

*Figure 1.* A visual depiction of the NTP' recursive computation graph construction, applied to a toy KB (top left). Dash-separated rectangles denote proof states (left: substitutions, right: proof score -generating neural network). All the non-FAIL proof states are aggregated to obtain the final proof success (depicted in Figure 2). Colours and indices on arrows correspond to the respective KB rule application.

most promising proof paths. In particular, we show that the problem of finding the facts in the KB that best explain a query can be reduced to a $k$-nearest neighbour problem, for which efficient exact and approximate solutions exist (Li et al., 2016). This enables us to apply NTPs to previously unreachable real-world datasets, such as WordNet.

## 2. Background

In NTPs, the neural network structure is built recursively, and its construction is defined in terms of *modules* similarly to dynamic neural module networks (Andreas et al., 2016). Each module, given a goal, a KB, and a current proof state as inputs, produces a list of new proof states, where the proof states are neural networks representing partial proof success scores.

**Unification Module.** In backward chaining, unification between two atoms is used for checking whether they can represent the same structure. In discrete unification, non-variable symbol are checked for equality, and the proof fails if the symbols differ. In NTPs, rather than comparing symbols, their *embedding representation* are compared by means of a Radial Basis Function (RBF) kernel. This allows matching different symbols with similar semantics,

such as matching relations like `grandFatherOf` and `grandpaOf`.

1. $\text{unify}_{\boldsymbol{\theta}}([\,],[\,],S)=S$
2. $\text{unify}_{\boldsymbol{\theta}}([\,],\mathrm{G},S)=\text{FAIL}$
3. $\text{unify}_{\boldsymbol{\theta}}(\mathrm{H},[\,],S)=\text{FAIL}$
4. $\text{unify}_{\boldsymbol{\theta}}(h::\mathrm{H},g::\mathrm{G},S)=\text{unify}_{\boldsymbol{\theta}}(\mathrm{H},\mathrm{G},S')$
   with $S'=(S'_\psi,S'_\rho)$ where:

$$S'_\psi = S_\psi \cup \left\{ \begin{array}{ll} \{h/g\} & \text{if } h \in \mathcal{V} \\ \{g/h\} & \text{if } g \in \mathcal{V}, h \notin \mathcal{V} \\ \varnothing & \text{otherwise} \end{array} \right\}$$

$$S'_\rho = \min\left(S_\rho, \left\{ \begin{array}{ll} \mathrm{k}(\boldsymbol{\theta}_{h:},\boldsymbol{\theta}_{g:}) & \text{if } h \notin \mathcal{V}, g \notin \mathcal{V} \\ 1 & \text{otherwise} \end{array} \right\}\right)$$

**OR Module.** This module attempts to apply rules in a KB. The name of this module stems from the fact that a KB can be seen as a large disjunction of rules and facts. In backward chaining reasoning systems, the OR module is used for unifying a goal with all facts and rules in a KB: if the goal unifies with the head of the rule, then a series of goals is derived from the body of such a rule. In NTPs,

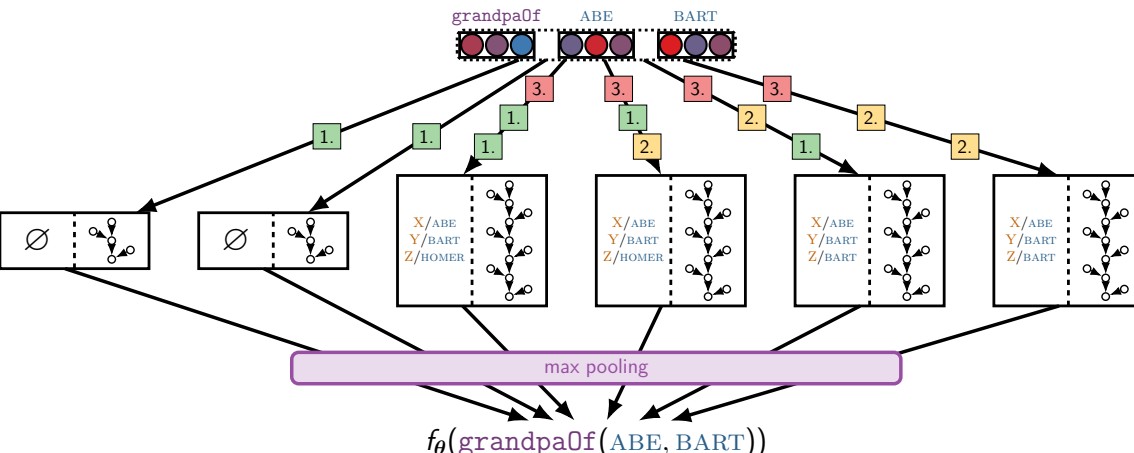

*Figure 2.* Depiction of the proof aggregation for the computation graph presented in Figure 1. Proof states resulting from the computation graph construction are all aggregated to obtain the final success score of proving a query.

we calculate the similarity between the rule and the facts via the `unify` operator. Upon calculating the continuous unification scores, OR calls AND to prove all sub-goals in the body of the rule.

$$\text{or}_{\boldsymbol{\theta}}^{\mathcal{K}}(\text{G}, d, S) = [S' \mid S' \in \text{and}_{\boldsymbol{\theta}}^{\mathcal{K}}(\text{B}, d,$$
$$\text{unify}_{\boldsymbol{\theta}}(\text{H}, \text{G}, S)), \text{H} :- \text{B} \in \mathcal{K}]$$

**AND Module.** This module is used for proving a conjunction of sub-goals derived from a rule body. It first applies substitutions to the first atom, which is afterwards proven by calling the OR module. Remaining sub-goals are proven by recursively calling the AND module.

1. $\text{and}_{\boldsymbol{\theta}}^{\mathcal{K}}(\_, \_, \text{FAIL}) = \text{FAIL}$

2. $\text{and}_{\boldsymbol{\theta}}^{\mathcal{K}}(\_, 0, \_) = \text{FAIL}$

3. $\text{and}_{\boldsymbol{\theta}}^{\mathcal{K}}([\,], \_, S) = S$

4. $\text{and}_{\boldsymbol{\theta}}^{\mathcal{K}}(\text{G} : \mathbb{G}, d, S) = [S'' \mid S'' \in \text{and}_{\boldsymbol{\theta}}^{\mathcal{K}}(\mathbb{G}, d, S')$
   for $S' \in \text{or}_{\boldsymbol{\theta}}^{\mathcal{K}}(\text{substitute}(\text{G}, S_{\psi}), d - 1, S)]$

For further details on NTPs and the particular implementation of these modules, see Rocktäschel and Riedel (2017)

After building all the proof states, NTPs define the final success score of proving a query as an argmax over all the generated valid proof scores (neural networks).

**Example 2.1.** Assume a KBs $\mathcal{K}$, composed of $|\mathcal{K}|$ facts and no rules, for brevity. Note that $|\mathcal{K}|$ can be impractical within the scope of NTP. For instance, Freebase (Bollacker et al., 2008) is composed of approximately 637 million facts, while YAGO3 (Mahdisoltani et al., 2015) is composed by approximately 9 million facts. Given a query $g \triangleq [\text{grandpaOf}, \text{ABE}, \text{BART}]$, NTP compares its embedding representation – given by the embedding vectors of $\text{grandpaOf}$, ABE, and BART – with the representation of each of the $|\mathcal{K}|$ facts.

The resulting proof score of $g$ is given by:

$$\max_{f \in \mathcal{K}} \text{unify}_{\boldsymbol{\theta}}(g, [f_p, f_s, f_o], (\varnothing, \rho))$$
$$= \max_{f \in \mathcal{K}} \min \big\{ \rho, \text{k}(\boldsymbol{\theta}_{\text{grandpaOf}:}, \boldsymbol{\theta}_{f_p:}), \qquad (1)$$
$$\text{k}(\boldsymbol{\theta}_{\text{ABE}:}, \boldsymbol{\theta}_{f_s:}), \text{k}(\boldsymbol{\theta}_{\text{BART}:}, \boldsymbol{\theta}_{f_o:}) \big\},$$

where $f \triangleq [f_p, f_s, f_o]$ is a fact in $\mathcal{K}$ denoting a relationship of type $f_p$ between $f_s$ and $f_o$, $\boldsymbol{\theta}_{s:}$ is the embedding representation of a symbol $s$, $\rho$ denotes the initial proof score, and $\text{k}(\cdot, \cdot)$ denotes the RBF kernel.

## 3. Nearest Neighbourhood Search

From Example 2.1, we can see that the inference problem can be reduced to a *nearest neighbour* search problem. Given a query $g$, the problem is finding the fact(s) in $\mathcal{K}$ that maximise the unification score. This represents a computational bottleneck, since it is very costly to find the exact nearest neighbour in high-dimensional Euclidean spaces, due to the curse of dimensionality (Indyk and Motwani, 1998). Exact methods are rarely more efficient than brute-force linear scan methods when the dimensionality is high (Ge et al., 2014; Malkov and Yashunin, 2016).

A practical solution consists in Approximate Nearest Neighbour Search (ANNS) algorithms, which relax the condition of the exact search by allowing a small number of mistakes. Several families of ANNS algorithms exist, such as Locality-Sensitive Hashing (LSH) (Andoni et al., 2015), Product Quantization (PQ) (Jégou et al., 2011), and Proximity Graphs (PGs) (Malkov et al., 2014). In this work we use Hierarchical Navigable Small World (HNSW) (Malkov and Yashunin, 2016; Boytsov and Naidan, 2013), a graph-based incremental ANNS structure which can offer much better logarithmic complexity scaling in comparison with other

*Table 1.* AUC-PR results on Countries and MRR and HITS@$m$ on Kinship, Nations, and UMLS.

| Dataset | Metric | | Model | |
|---|---|---|---|---|
| | | | NTP | NTP 2.0 ($k = 1$) |
| **Countries** | S1 | AUC-PR | $90.83 \pm 15.4$ | $\mathbf{97.04 \pm 4.47}$ |
| | S2 | AUC-PR | $87.40 \pm 11.7$ | $\mathbf{90.92 \pm 4.44}$ |
| | S3 | AUC-PR | $56.68 \pm 17.6$ | $\mathbf{85.55 \pm 7.10}$ |
| **Kinship** | | MRR | 0.60 | **0.65** |
| | | HITS@1 | 0.48 | **0.57** |
| | | HITS@3 | **0.70** | 0.69 |
| | | HITS@10 | 0.78 | **0.81** |
| **Nations** | | MRR | 0.75 | **0.81** |
| | | HITS@1 | 0.62 | **0.73** |
| | | HITS@3 | **0.86** | 0.83 |
| | | HITS@10 | **0.99** | **0.99** |
| **UMLS** | | MRR | **0.88** | 0.76 |
| | | HITS@1 | **0.82** | 0.68 |
| | | HITS@3 | **0.92** | 0.81 |
| | | HITS@10 | **0.97** | 0.88 |

*Table 2.* Rules induced on WordNet, with a confidence above $0.5$.

| Confidence | Rule |
|---|---|
| 0.584 | _domain_topic(X, Y) :-_domain_topic(Y, X) |
| 0.786 | _part_of(X, Y) :-_domain_region(Y, X) |
| 0.929 | _similar_to(X, Y) :-_domain_topic(Y, X) |
| 0.943 | _synset_domain_topic(X, Y) :-_domain_topic(Y, X) |
| 0.998 | _has_part(X, Y) :-_similar_to(Y, X) |
| 0.995 | _member_meronym(X, Y) :-_member_holonym(Y, X) |
| 0.904 | _domain_topic(X, Y) :-_has_part(Y, X) |
| 0.814 | _member_meronym(X, Y) :-_member_holonym(Y, X) |
| 0.888 | _part_of(X, Y) :-_domain_topic(Y, X) |
| 0.996 | _member_holonym(X, Y) :-_member_meronym(Y, X) |
| 0.877 | _part_of(X, Y) :-_domain_topic(Y, X) |
| 0.945 | _synset_domain_topic(X, Y) :-_domain_region(Y, X) |
| 0.879 | _part_of(X, Y) :-_domain_topic(Y, X) |
| 0.926 | _domain_topic(X, Y) :-_domain_topic(Y, X) |
| 0.995 | _has_instance(X, Y) :-_type_of(Y, X) |
| 0.996 | _type_of(X, Y) :-_has_instance(Y, X) |

approaches.

## 4. Related Work

Many machine learning methods rely on efficient nearest neighbour search for solving specific sub-problems. Given the computational complexity of nearest neighbour search, approximate methods, driven by advanced index structures, hash or even graph-based approaches are used to speed up the bottleneck of costly comparison. These algorithms have been used to speed up various sorts of machine learning models, from mixture model clustering (Moore, 1999), case-based reasoning (Wess et al., 1993) to Gaussian process regression (Shen et al., 2006), among others.

In kind, the most similar work to ours is the work of Rae et al. (2016) who apply approximate nearest neighbours to speed up Memory-Augmented neural networks. Similarly to our work, they apply ANNS to query the external memory (in our case the KB memory) for $k$ closest words. They present drastic savings in speed and memory usage. Though as of this moment, our speed savings are not as drastic, the memory savings we achieve are sufficient so that we can train on WordNet, a dataset previously considered out of reach of NTPs.

## 5. Experiments

We compared results obtained by our model, which we refer to as NTP 2.0, with those obtained by the original NTP proposed by Rocktäschel and Riedel (2017). Results on several smaller datasets – namely Countries, Nations, Kinship, and UMLS – are shown in Table 1. When unifying goals with facts in the KB, for each goal, we use ANNS for re-

trieving the $k$ most similar (in embedding space) facts, and use those for computing the final proof scores. We report results for $k = 1$, as we did not notice sensible differences for $k \in \{2, 5, 10\}$. However, we noticed sensible improvements in the case of Countries, and an overall decrease in performance in UMLS. One possible explanation is that ANNS (with $k = 1$), due to its inherently approximate nature, does not always retrieve the closest fact(s) exactly: this behaviour may be a problem in some datasets where exact nearest neighbour search is crucial for correctly answering queries. On the other hand, it may even improve training in other datasets since gradients would flow through proof paths that would not be considered otherwise.

We also evaluated NTP 2.0 on WordNet (Miller, 1995), a KB encoding lexical knowledge about the English language. In particular, we use the WordNet used by Socher et al. (2013) for their experiments. This dataset is significantly larger than the other datasets used by Rocktäschel and Riedel (2017) – it is composed by 38.696 entities, 11 relations, and the training set is composed by 112,581 facts.

In WordNet, the accuracies on the validation and test sets were 65.29% and 65.72%, respectively – which is on par with the Distance Model, a Neural Link Predictor discussed by Socher et al. (2013), which achieves a test accuracy of 68.3%. However, we did not consider a full hyper-parameter sweep, and did not regularise the model using Neural Link Predictors, which sensibly improves NTPs' predictive accuracy (Rocktäschel and Riedel, 2017). A subset of the induced rules is shown in Table 2.

## 6. Conclusion

We proposed a way to sensibly scale up NTPs by reducing parts of their inference steps to ANNS problems, for which very efficient and scalable solutions exist in the literature.

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
