# OpenReview forum: "Towards Neural Theorem Proving at Scale"
_ICML.cc/2018/Workshop/NAMPI — NAMPI 2018_

### Review · AnonReviewer1 · 2018-06-14
**Nice trick to avoid complexity blowup**

**Rating:** 7
**Confidence:** 4

**Review:**

The paper describes an extension to the Neural Theorem Proving work that allows scaling to substantially larger knowledge bases. The core insight is to use an approximate nearest neighbour search (in embedding space) to filter the set of facts that are used in the full logic programming-like routine. By aggressively filtering this to the most likely candidates, the system only needs to consider a small number of potential proofs.

Overall, this seems like a nice idea, though not breakthrough work. The paper has a few oddities in writing (see below) but is mostly readable.  I recommend acceptance to the workshop, but encourage the author to consider the following two points:
(1) The writing seems a bit chaotic, and the core contribution of the paper (filtering candidates for unification attempts using ANNS) is only made explicit in the experiments section. Highlighting this, and maybe replacing/extending Ex. 2.1 to explicitly illustrate how the method is changed, would help.
(2) The experiments could be more informative, for example by showing how the system would work on tasks that require deeper proofs (i.e., of a few steps), and plotting the loss of precision due to the filtering procedure. Intuitively, the proposed method feels like beam search (in that only promising branches of the full proof tree are explored), and so you would expect that it does in deeper trees. I would also be curious to see other related extensions (i.e., branch-and-bound-like techniques during the exploration) be evaluated.

Minor things:
- l104: "variable symbol are" -> "variable symbols are"
- l106: "embedding representation are" -> "embedding representations are"
- Item 4 of unify definition: S_\psi, S_\rho are undefined, could be done with "with $S' = (S'_\psi, S'_\rho)$ and $S = (S_\psi, S_\rho)$" at top.
- Item 4 of unify definition: \theta_{h:}, \theta_{g:} are undefined syntax (I assume you mean the learned embedding of h/g?); maybe make this an explicit emb(h), emb(g)?
- l133 (Definition of or): The line break in the call of and() is confusing, maybe break after it?
- l147: This is not python, the "for" is not required (and not used in the other defs)

---

### Review · AnonReviewer2 · 2018-06-21
**Good paper**

**Rating:** 7
**Confidence:** 3

**Review:**

This paper describes a method for scaling up the Neural Theorem Prover so that it can work on much larger knowledge bases (e.g. WordNet). They optimise the unification step between a query and a set K of facts by using ANN to find a K' of facts that will maximise the unification score, where |K'| << |K|.

This seems like a sensible approach, and it is impressive that the authors are able to scale NTP so that it can work on larger databases. It is also reassuring that their results on smaller databases are not significantly worse than the original NTP. In some cases, the results for NTP 2.0 are actually better, which is prima facie surprising. The authors attempt an explanation of this fact: "gradients would flow through proof paths that would not be considered otherwise." I do not fully understand this explanation.

One thing that would, I believe, improve the paper is a formula describing the space and time usage of the original NTP as a function of the set K of facts, the number R of rules, and the number N of steps of inference. I want to know how many unifications are performed, as a function of K, R, and N. I want to know the size of the network, as a function of K, R, and N. My understanding is that, because of the exponential memory usage, the original NTP was very restricted in the number N of steps of inference that can be performed. How much more can N be increased in NTP 2.0?

Anyway, this is a nice, well-written, clear paper. I would recommend focusing a little less space on the details of the original NTP, and a bit more space on the time/space analysis of NTP, and more space on the details of how you applied ANN to reduce K.

---

### Decision · ~NAMPI_Admin1 · 2018-06-28
**Paper9 Final Decision**

Accept